# Glucagon-Like Peptide-1 Receptor Agonists in Patients with Type 2 Diabetes: Prescription According to Reimbursement Constraints and Guideline Recommendations in Catalonia

**DOI:** 10.3390/jcm8091389

**Published:** 2019-09-05

**Authors:** Josep Franch-Nadal, Manel Mata-Cases, Emilio Ortega, Jordi Real, Mònica Gratacòs, Bogdan Vlacho, Joan Antoni Vallés, Dídac Mauricio

**Affiliations:** 1DAP-Cat group, Unitat de Suport a la Recerca Barcelona, Fundació Institut Universitari per a la Recerca a l’Atenció Primària de Salut Jordi Gol i Gurina (IDIAPJGol), 08007 Barcelona, Spain (J.F-N) (M.M-C.) (J.R.) (M.G.) (B.V) (J.A.V.); 2CIBER of Diabetes and Associated Metabolic Diseases (CIBERDEM), Instituto de Salud Carlos III (ISCIII), 28029 Madrid, Spain; 3Primary Health Care Center Raval Sud, Gerència d’Atenció Primaria, Institut Català de la Salut, 08028 Barcelona, Spain; 4Primary Health Care Center La Mina, Gerència d’Àmbit d’Atenció Primària Barcelona Ciutat, Institut Català de la Salut, Sant Adrià de Besòs, 08930 Barcelona, Spain; 5Department of Endocrinology and Nutrition, Institut d’Investigacions Biomèdiques August Pi i Suñer, Hospital Clinic, 08036 Barcelona, Spain; 6CIBER of Physiopathology of Obesity and Nutrition (CIBEROBN), Instituto de Salud Carlos III (ISCIII), 28029 Madrid, Spain; 7Drug Area, Gerència d’Atenció Primaria, Institut Català de la Salut, 08028 Barcelona, Spain; 8Department of Endocrinology and Nutrition, Hospital de la Santa Creu i Sant Pau, Autonomous Universtity of Barcelona, 08041 Barcelona, Spain

**Keywords:** GLP-1 analogue, type 2 diabetes mellitus, primary care, observational study, glycaemic control

## Abstract

To assess the clinical characteristics, the prescription pattern of GLP-1 receptor agonists (GLP-1RA) users, and HbA1c and weight change, we retrospectively assessed patients with type 2 diabetes by initiating GLP-1RA as an add-on to the standard of care in Catalonia. The mean change from the baseline in glycated hemoglobin (HbA1c) and weight at 6 and 12 months of therapy was calculated, and we assessed the predictors of the HbA1c reduction of ≥1% and/or the weight reduction of ≥3% as recommended by the Catalan Health Service. In 2854 patients who initiated a GLP-1RA during 2014 and 2015, the overall mean HbA1c values were reduced from the baseline by −0.84% (SD = 1.66) (−9.2 mmol/mol) and lost on average 2.73 kg (SD = 6.2). About 44% percent of patients decreased their HbA1c by ≥1%; 44% decreased their weight by ≥3%; and only 22% met both of them together. The odds of achieving a reduction of ≥1% in initial HbA1c were two-fold higher for patients with higher baseline levels, and the likelihood of a reduction of ≥3% in the initial weight was associated with a higher BMI at the baseline, but they were independent of each other. The composite outcome (target 1% HbA1c reduction and 3% weight loss) to evaluate both the GLP-1RA clinical benefit and treatment withdrawal should be judged from a patient-centered approach.

## 1. Introduction

Type 2 diabetes (T2D) is characterized by a progressive decline in the β-cell function with a parallel reduction in insulin secretion that gradually raises hyperglycemia despite glucose-lowering medication [1,2]. Indeed, many patients initially treated with oral antidiabetic drugs (OADs), such as metformin (Met) of sulphonylureas (SU), eventually need to intensify therapy to maintain normal blood glucose levels. Typically, this is accomplished through the addition of a second or third OAD or switching to injectable agents, at present insulin and glucagon-like peptide-1 (GLP-1) receptor agonists (RAs) [3,4,5].

GLP-1 is a naturally occurring intestinal hormone that potentiates insulin secretion and decreases glucagon release, it delays gastric emptying, increases satiety, and protects β-cell mass [6,7]. Both in the European Union and the United States, class GLP-1RAs include agents administered twice daily (short-acting; i.e., exenatide), administered once daily (intermediate-acting; i.e., liraglutide and lixisenatide), and administered once weekly (long-acting; i.e., exenatide extended-release, dulaglutide, and albiglutide [not currently marketed or distributed]) [8,9].

Although effective as monotherapy, GLP-1RAs are mostly recommended in clinical guidelines as dual therapy with metformin and as triple therapy in combination with Met and a SU, a thiazolidinedione, or insulin [10,11,12]. In clinical practice, GLP-1RAs are mainly used as an alternative to the addition of basal insulin or to the intensification of insulin with prandial insulin in patients already in basal insulin [13]. This is because insulin therapy involves an increased risk of severe hypoglycemia, a certain increase in weight, the need for regular self-monitoring to adjust the dose, and continuous professional guidance [14]. Besides a reduced regimen complexity (no need for self-management and dose titration), randomized clinical trials (RCTs) have proven that GLP-1RAs are effective at reducing glycated hemoglobin (HbA1c) while reducing the risk of hypoglycemic events [9,15,16]. Moreover, they add potential benefits, such as an induction of weight loss, a reduction of blood pressure and lipids, an improvement in the β-cell function, and also a reduction of cardiovascular events [9,15,16,17,18]. The 2019 ADA/EASD Consensus Report considers GLP-1Ras the preferable first injection therapy in T2DM [5]. Finally, although the extent of the improvement in metabolic control varies between observational studies, they also show that GLP-1RAs are as effective and safe in real-world clinical practice as in RCTs [19].

The experience regarding long-term safety and effectiveness of GLP-1RAs is limited because they were marketed only about ten years ago, and the cost of this class of drugs is substantial compared to other agents [20]. For these reasons, guidelines from some European countries, including the United Kingdom (National Institute for Health and Clinical Excellence—NICE) and Catalonia (Catalan Health Service—CatSalut), recommend to discontinue treatment if no beneficial response (defined as a reduction of at least 1.0 percentage point in HbA1c [11 mmol/mol] and 3% in initial weight) has been observed after six months of therapy initiation [21,22]. In Spain, GLP-1RA are only reimbursed in obese patients (body mass index [BMI] ≥30 kg/m^2^) [21,22]. Finally, CatSalut recommends prioritizing these drugs as the third step in patients with BMI ≥35kg/m^2^. Similarly, the NICE recommends prioritizing them to subjects with BMI ≥35 kg/m^2^ and obesity-related medical problems and, also, to those non-obese for whom weight loss is desirable to prevent obesity-related comorbidities or for whom insulin therapy is not appropriate because of occupational repercussions [21].

The primary objective of the current study was to retrospectively assess the characteristics of patients with T2D who initiated GLP-1RA in primary health care in Catalonia (Spain). We evaluated glycemic and weight control after 6 and 12 months of therapy to determine whether the treatment was beneficial to reducing glycemia and weight according to the national clinical guideline (CatSalut) and health policy requirements for reimbursement. Moreover, we examined whether the prescription was compliant with the instructions on the European public assessment report (EPAR) and summary of product characteristics (SmPC) and reported the occurrence of serious adverse events.

## 2. Experimental Section

### 2.1. Study Population

The present study is a retrospective analysis using an electronic database of computerized primary care records (SIDIAP; System for the Development of Research in Primary Care) available for researchers [23,24]. The SIDIAP database contains demographic and clinical information, laboratory test results, prescriptions, and referrals recorded by >3500 general practitioners (GPs) through a common software (eCAP) used by all 335 primary care centers (PCCs) managed by the Catalan Health Institute (ICS). The ICS provides health care to about 5.6 million people, representing approximately 74% and 12% of the total population of Catalonia and Spain, respectively. Moreover, the SIDIAP database incorporates data on dispensed treatments extracted from pharmacy-invoicing records provided by the Catalan Health Service (CatSalut). Anonymized data from the SIDIAP database were obtained that covered the period between January 2014 and December 2015.

The study was approved by the Ethics Committee of the Primary Health Care University Research Institute (IDIAP) Jordi Gol, Barcelona (P17/018).

### 2.2. Inclusion and Exclusion Criteria

The study enrolled patients aged 31 to 90 years with an ICD-10 (International Classification of Diseases) diagnose code for T2D (i.e., E.11 or E11.0-E11.9). All patients had to have been prescribed one agent of the GLP-1RA drug class for the first time (index date) between 1 January 2014 and 31 December 2015 and followed until December 2016. The approved GLP-1-RAs included in this study were liraglutide and lixisenatide once a day and exenatide (twice a day [BD] or once per week [QW]). Albiglutide and dulaglutide were not analyzed because they were launched in 2015 in Spain and were very rarely prescribed (only 42 patients). Patients with a diagnosis code for type 1 diabetes, gestational diabetes or secondary diabetes (CIE-10: E10; O24.4; or E09) were excluded from the study.

### 2.3. Study Variables

The following baseline variables were extracted from the electronic clinical records: (1) Age and gender; (2) variables related to the T2D: Year of diagnosis, fasting plasma glucose (FPG) and HbA1c values previous to the introduction of GLP-1RA therapy (baseline or index date), at six months of treatment (the time window was three to six months after the initiation), and also at 12 months of treatment (the time window was from 6 to 12 months after the initiation, taking the closest value to 12 months); (3) a previous presence of risk factors and long-term complications (previous three months as a maximum) and at follow up (6 and 12 months after initiation), including: Weight, body mass index (BMI), total cholesterol, LDL-cholesterol, HDL cholesterol, triglycerides, systolic and diastolic blood pressure (SBP and DBP), heart rate, smoking status, creatinin, estimated glomerular filtration rate (eGFR) using the Chronic Kidney Disease Epidemiology Collaboration (CKD-EPI) equation, presence of microvascular complications (diabetic retinopathy and nephropathy), presence of macrovascular complications (stroke, peripheral artery disease and ischemic heart disease), heart failure), history of severe gastrointestinal disease (i.e., gastroparesis, inflammatory bowel disease, acute or chronic pancreatitis, and pancreatic cancer), and thyroid cancer during the study period, and use of antiemetic agents (i.e., metoclopramide or clebopride); (4) previous oral or insulin antidiabetic treatments prescribed on the index date and during the study period trough Anatomical Therapeutic Chemical (ATC)/Defined Daily Doses (DDD) codes; and (5) ATC/DDD codes for the type of GLP-1RA prescribed, the date of the first prescription, and the date of discontinuation if it was suspended or changed for another agent of the same class. As daily and weekly exenatide share the same ATC/DDD code, it was not possible to identify the prescription of these two pharmaceutical formulations separately.

### 2.4. Statistical Methods

Descriptive analyses were summarized by the mean and standard deviation for continuous variables (median and interquartile range if non-normally distributed) and absolute frequency and percentages for categorical variables. We calculated the proportion of patients that were prescribed a GLP-1RA in accordance with: (1) The GLP-1RAs EPAR and label, namely: if HbA1c ≥ 7%, treated with other glucose-lowering agents, not used in patients with end-stage or severe chronic kidney disease (CDK), or used with caution—or adjusting the dose—in moderate CDK; (2) the local health policy requirements for reimbursement (HPR; i.e., HbA1c ≥ 7%,%, previously treated with other glucose-lowering agents, and a BMI ≥30 kg/m^2^); and (3) the NICE and national, local guideline (CatSalut) targets to consider the treatment as cost-effective—defined as a reduction of at least 1.0% (11 mmol/mol) in HbA1c and a weight loss of at least 3% of the initial body weight at 6 and at 12 months after treatment initiation [21,22]. The mean change from the baseline and the significance of the decrease in HbA1c values (overall and stratified by levels <7%, 7.0–7.9%, 8.0–9.9, and ≥10%) and weight after 6 and 12 months of treatment was assessed through matched-pairs t-tests. Moreover, we estimated variables predicting, after 6 and 12 months of treatment initiation, HbA1c reduction ≥1%, weight reduction ≥3%, one or the other target, or both of them together. For this, we used multivariate logistic regression models using the Enter method, adjusting for the baseline characteristics with a significance <0.1 in the univariate analysis. Parameter estimates were expressed as odds ratios (OR) and 95% confidence intervals (95% CI). All hypothesis contrasts were bi-directional, and the statistical significance level was set at 0.05. Finally, we calculated the incidence rate and cumulative incidence of acute and chronic pancreatitis, and pancreas and thyroid cancer that occurred during the study follow-up. We also calculated the treatment persistence, defined as the mean time with the same treatment, and the adherence to treatment, defined as 80% of the prescriptions picked up at the pharmacy. All statistical calculations were performed using StataCorp 2009 (Stata Statistical Software: Release 12, College Station, StataCorp, 4905 Lakeway Drive, TX, USA.).

## 3. Results

### 3.1. Patients’ Characteristics

A total of 2854 patients with T2D (50.6% male) initiated therapy with a GLP-1RA and were followed for a mean of 23.9 months (SD = 7.2). The baseline characteristics are shown in Table 1. The median age was 59.5 years (SD = 9.8), and 42.0% of patients had a diabetes duration longer than 10 years. Treatment with GLP-1RAs was initiated with a mean HbA1c value of 8.74% (SD = 1.6) and a mean basal BMI of 37.1 kg/m^2^ (SD = 5.9).

Almost half of the patients (45.7%) initiating a GLP-1RA were treated with ≥2 antidiabetic agents (including insulin). The most frequently used drug was metformin (81.1% of patients), and 52% of patients were on insulin treatment. The most commonly prescribed GLP-1RA was liraglutide (56%), followed by exenatide (BID or QW; 23.8%) and lixisenatide (20.2%).

### 3.2. Adequacy to GLP-1RAs SmPC and Local Health Policy Requirements for Reimbursement

Based on the CatSalut guideline (and also the NICE guideline), GLP-1RAs are an option if given in combination with other antidiabetics to patients with suboptimal glycemic control (HbA1c level ≥ 7% and > 7.5%, respectively) and a BMI ≥35 kg/m^2^ (and only reimbursed in Catalonia if BMI is ≥30 kg/m^2^) [21,22]. The fulfilment of all three conditions occurred in 79.5% of patients. In the study cohort, 12.3% of patients had previous HbA1c values of <7%; 8.7% had a baseline BMI of <30 kg/m^2^; and in 3.3% of cases the GLP-1RA was prescribed as monotherapy (i.e., no previous pharmacological treatment). Moreover, and based on SmPC, GLP-1RAs must be used with caution in patients with renal impairment (eGFR < 45 mL/min/1.73 m^2^), but they were prescribed to subjects below this cut-off in 0.5% of cases.

### 3.3. Treatment Response after Six and Twelve Months of GLP-1RA Initiation

After six months of GLP-1RA treatment initiation, the overall mean HbA1c values were reduced by 0.84% (SD = 1.66) (−9.2 mmol/mol), and patients lost in average 2.73 kg (SD = 6.44) (Table 2). The largest reductions in HbA1c levels were observed among patients with baseline values of >10% (mean reduction −2.20% vs. 0.87 if previously HbA1c was 7–10%; *p* < 0.0001), and the greatest reductions in weight were experienced by those with an initial BMI >35 kg/m^2^ (mean reduction −3.65 kg vs. −1.71 if the baseline BMI was 30–35 kg/m^2^ and 0.27 kg if the BMI was <30–35 kg/m^2^; *p* < 0.0001) (Appendix A). Moreover, we also observed reductions in blood pressure, cholesterol levels, triglycerides, and eGFR (Table 2). These changes remained after 6–12 months of treatment, with a mean reduction of 0.88% in HbA1c values and 2.86 kg of the initial weight (Table 2). Similarly, the greatest benefit of treatment was as well observed after 12 months of GLP-1RA initiation among those with the poorest initial glycemic and the highest BMI (Appendix A).

The mean changes from the baseline in HbA1c and triglycerides were significantly larger among patients who fulfilled both the prescription instructions and the local health policy for reimbursement at both 6 and 12 months of treatment (Table 2). In addition, the changes in BMI and total cholesterol at 12 months were significantly greater among patients fulfilling the local health prescription policy rules, which include the criterion of an initial BMI of ≥30 kg/m^2^. For all the other variables (e.g., blood pressure, kidney function, or weight), there were no significant differences between patients prescribed GLP-1RAs according to the package label or health policy recommendations at any endpoint.

### 3.4. Assessment of Beneficial Response

The CatSalut (Catalonia) and NICE guidelines (United Kingdom) require a reduction of at least 1% in HbA1c and of 3% in initial weight after six months of GLP-1RAs initiation to consider them as beneficial and to continue treatment [21,22]. Overall, more than half of the patients (57.1%) met at least one goal after six months of treatment, but less than a quarter achieved both of them together (22.4%) (Figure 1 and Appendix A). As for single goals, the percentage of patients achieving only a 1% HbA1c reduction or a weight reduction of only ≥3% was similar (43.1% and 45.0%, respectively).

The proportion of patients experiencing a beneficial metabolic response varied depending on whether the prescription was made according to label instructions (SmPC) and to the local health policy for reimbursement (HPR). Among patients fulfilling the SmPC and HPR prerequisites, the probabilities of achieving an HbA1c reduction ≥1% alone and/or together with a weight reduction of ≥3% were significantly higher (*p* < 0.001) than among patients who did not. However, although a baseline BMI of >30 kg/m^2^ is a prerequisite in the case of the HPR, the probability to lose ≥3% of the initial weight as a single goal were similar despite an initial BMI of ≥30 kg/m^2^ (Figure 1B and Appendix A). After 12 months of treatment, the proportions were similar to the ones observed at six months (Appendix A).

Moreover, the probability of achieving an HbA1c reduction >1% alone and/or together with a weight reduction of ≥3% were significantly higher (*p* < 0.001) in patients with the highest baseline weight levels (i.e., >10%) (Figure 2A and Appendix A) and also among patients with the highest baseline BMI (i.e., >35 Kg/m^2^) (Figure 2B and Appendix A). After 12 months of treatment, the proportions were comparable to the ones observed at six months (Appendix A).

### 3.5. Medication Persistence and Adherence

The mean time that patients continued the treatment (persistence) when the first GLP-1RA was prescribed was 23.9 months (SD = 7.2). At the end of follow-up, 48% of patients had been adherent to medication, defined as picking up 80% of the prescriptions at the pharmacy.

### 3.6. Adverse Events

After 12 months of treatment, five cases of pancreatic cancer (0.18%), 11 cases of acute or chronic pancreatitis (0.39%), and two cases of thyroid cancer (0.07%) were recorded (Appendix A). Liraglutide was the agent associated with the highest frequency of pancreatic events (0.5% vs. 0.2% with lixisenatide and 0.3% with exenatide), while there were no substantial differences between exenatide and lixisenatide.

### 3.7. Variables Associated with Beneficial HbA1c Reduction and Weight Loss after Therapy with GLP-1RAs

The results of the variables identified in the multivariate regression analyses as associated with the likelihood to attain beneficial metabolic goals after six months and 6–12 months of treatment are shown in Table 3. Complete data are provided in Appendix A. 

After six months, the odds of achieving a reduction ≥1% in HbA1c were two-fold higher for patients with higher baseline levels (*p* < 0.001) and less likely among those with T2D duration 10–20 years (*p* = 0.036). The only variable associated with the likelihood of a reduction of ≥3% in initial weight was the baseline BMI (*p* = 0.014). The probability of achieving both a reduction in HbA1c ≥1% and >3% weight loss was higher among those with high baseline HbA1c values (*p* < 0.001) and with high baseline BMI (*p* = 0.032). When considering the possibility of one goal, the other, or both, the odds were higher among patients with high baseline HbA1c and BMI values (*p* < 0.001 and *p* = 0.043, respectively) and those fulfilling the HPR requirements (*p* = 0.031), but lower if the diabetes duration was 10–20 years (*p* = 0.031).

After 12 months of treatment, there were few variations regarding the variables associated with the goals’ achievement (Table 3 and Appendix A), except that: (1) The male gender was as well associated with less likelihood of reducing weight ≥3% (*p* = 0.036); (2) a T2D duration of 10–20 years was no longer associated with the lower likelihood of achieving a reduction in HbA1c ≥1% (or ≥3% reduction weight), but those with T2D duration longer than 20 years had an almost three-fold higher probability of achieving both outcomes together (*p* = 0.042); (3) DBP and eGFR were, as well, associated with higher odds of achieving both outcomes (*p* = 0.018 and *p* = 0.042, respectively); and (4) prescription according to HPR had no influence on the odds of achieving any of the studied outcomes.

## 4. Discussion

The present retrospective study shows that in real-world clinical practice in Catalonia, the profile of the patient that is prescribed a GLP-1Ra is a middle-aged person (about 60 years) with diabetes lasting for approximately 10 years, poor glycemic control (HbA1c > 8%), and obese (BMI > 35 kg/m^2^), but otherwise with an overall good control of other risk factors (e.g., lipid profile, hypertension and chronic kidney disease). After six months of therapy, the baseline HbA1c values were reduced by a mean of 0.84% (9 mmol/mol) and patients lost on average 2.73 kg, changes that were maintained after 12 months of medication intensification.

The magnitude of the observed reductions in HbA1c and initial weight is in line with that reported in systematic reviews and meta-analyses of RCTs (between 0.4–1.8% and 1.2–3.9 kg, respectively), and also with that from observational and retrospective studies (between 0.5–1.4% and 3.1–6.5 kg, respectively) [8,16,19,25]. Also, as previously reported, the largest reductions in HbA1c levels were observed among patients with the worst baseline glycemic control and the largest weight reductions among patients with severe obesity [26,27,28,29,30].

To our knowledge, there are no currently available data on inappropriate use or medication errors with GLP-1RAs, and little data exist on its use in special populations. In the present study, 20.5% of patients who initiated GLP-1RA therapy had a baseline HbA1c value <7%, their BMI was ≤30 kg/m^2^, or were prescribed as monotherapy, which is not in line with our local clinical guideline initiation criteria. This is consistent with audits conducted in the primary care setting, where 20–50% of patients deviated from the HbA1c and BMI NICE recommendations (i.e., initiate as dual/triple therapy, and if HbA1c > 7.5% and BMI > 35 kg/m^2^) [31,32,33,34]. In particular, 12.3% of our patients were prescribed a GP-1RA despite a good glycemic control, which is in line with a real-world study conducted in Spain reporting that 14% of patients initiating GLP-1RAs had a baseline HbA1c value <7% [35]. However, it is lower than in a study conducted in the United States where, prior to liraglutide initiation, about a third of patients had HbA1c values within the target range [36]. This indicates that, in a non-negligible number of cases, GLP-1RAs are probably given to reduce weight rather than to control hyperglycemia. Indeed, 9% of patients in our cohort initiated a GLP-1RA with a BMI < 30 kg/m^2^ irrespective of the administrative restrictions. Although GLPRAs are equally effective in patients with a BMI below 30 kg/m^2^ [37], they are not reimbursed because of high cost and budget constraints. RCTs and some studies conducted in clinical practice have shown that GLP-1RAs, as a drug class, have a significant impact on weight loss in non-diabetic subjects, although in the latter case side-effects and discontinuation rates were higher than in controlled trials [38,39,40]. This calls into question whether treatment with GLP-1RAs should be considered an acceptable practice in a subset of obese patients (e.g., those for whom bariatric surgery other long-term pharmacological treatments are not suitable, or have obesity-related problems), as accepted in the case of the NICE guidelines.

Renal insufficiency is not a contraindication but rather a recommendation not to use these drugs due to lack of experience and this has been changed over time. Initially, these drugs were not recommended for patients with eGFR values below 30 mL/min (daily exenatide, liraglutide, and lixisenatide), and with eGFR < 50 mL/min for weekly exenatide. Later on, the lower limit for eGFR was changed to 15 mL/min for liraglutide. Therefore, we considered, by internal criteria, eGFR < 45 mL/min as a not recommended prescription. In addition, in our cohort, only 10 (0.5%) patients initiated GLP-1RA therapy in spite of having a compromised renal function (eGFR < 30 mL/min/1.73 m^2^), a proportion higher than the 9.3% reported by a study conducted in primary care in the United States [41]. This is a breach of the prescription instructions, which recommend not to use them in patients with end-stage or severe CDK and to use them with caution—or adjust the dose—in moderate CDK. In our study, the overall mean eGFR decreased by 1.63 and 1.2 units at six months and 12 months, respectively. However, evidence from available studies indicate that patients with T2D and CKD have lower probabilities of developing or experiencing a further deterioration in their renal function when treated with GLP-1RAs than with placebo or other glucose-lowering agents [41,42]. This effect could be attributed to the direct renoprotective properties of GLP-1RA receptors’ activation in the kidney and to the added indirect benefits in glycemic control, hypertension, and body weight [42,43]. Based on these results, some authors advocate that GLP-1RAs might be underused in this special population [41].

Less than a quarter of patients (22.4%) experienced a beneficial metabolic response according to the six months NICE and CatSalut continuation criteria (HbA1c ≥ 1% reduction with ≥3% weight loss), which is in line with the 25%–34% reported by other studies conducted in primary care [29,30,31,32,33,34,44]. This would argue for an independent assessment of the reduction in weight and glycemia in the decision to continue GLP-1RAs therapy, which is further supported by the results of the multivariate analysis. The most important predictor of the reduction of ≥1% in HbA1c was a high baseline value, as previously reported by both observational studies and RCTs [45,46,47,48,49], and the odds were independent of the baseline BMI. Besides, a high initial BMI is the most important determinant of a weight reduction of ≥3%, which is also in line with other studies [50], and the likelihood was in turn independent of the baseline HbA1c.

The independent likelihood of achieving a beneficial change in glycemia and weight loss has been previously observed [26,27,45,49,50,51], and the weak or absent correlation between the magnitude in HbA1c and weight loss is not new [26,27,34,50,51]. The reasons for this could relate to the pleiotropic actions and distinct physiopathological pathways triggering each of the effects [43], and the patient-to-patient variability in the degree of improvement in glycemia and weight could be partially attributed to an underlying differential biological response. Based on this, it has been called into question whether it is acceptable to stop treatment in patients who have not met both criteria. Accordingly, some have proposed that patients achieving a significant HbA1c reduction, but not substantial weight loss, should be allowed to continue treatment [31,50,52,53]. The opposite situation should be also considered: In patients for whom weight reduction is not an easy goal to achieve or maintain, a significant reduction of ≥3% could be considered as beneficial despite a concomitant, less than “satisfactory” reduction in HbA1c levels, as this would, in turn, be associated with an improvement in the vascular risk [50,53].

The strengths of our study include the large sample size coming from a primary health care setting, which reflects routine clinical practice, although it may not be entirely representative of other areas inside or outside Spain with different health care systems or prescription policies. The results of our study should be interpreted keeping in mind potential limitations. Firstly, the retrospective observational design and the use of electronic clinical records entails the lack of randomization, resulting in a numerical imbalance between the different GLP-1 cohorts (liraglutide was used by more than half of the patients). Secondly, changes in HbA1c and weight from the baseline were calculated for subjects with data at all timepoints, which may have introduced bias. Thirdly, we were unable to investigate reasons for treatment discontinuation during the study (primary vs. secondary failure) and of particular adverse events, such as the incidence of hypoglycemia, as they are in general not accurately reported to general practitioners and rarely recorded in the database. Moreover, we did not record the occurrence of some serious health conditions, such as abnormal thyroid function, gastrointestinal disorders, malnutrition, or chronic inflammations, among others, which could have influenced BMI or be, as well, a cause of discontinuation. Finally, the ATC code for exenatide is the same for the BID and QW formulations, which prevented a homogenous follow-up and individual analyses by drug. However, based on other data from the CatSalut Pharmacy Register in the years of the present study, the vast majority of them (90%) were weekly formulations. Besides, we could not differentiate between doses of liraglutide, although the highest dose (1.8 µg/day) has been reported as more efficacious and a factor predicting the reduction of HbA1c ≥ 1.0% [49].

## 5. Conclusions

The present findings indicate that HbA1c and weight reductions in patients initiating GLP-1RA in routine clinical practice were in the expected range. However, the composite outcome of a beneficial effect in HbA1c coupled with a significant weight loss as stated by NICE and CatSalut was hard to reach for approximately 80% of the patients and the recommendation to stop the treatment should be regarded with caution, indeed suggesting that treatment response should be judged from an individualized and patient-centered approach.

## Figures and Tables

**Figure 1 jcm-08-01389-f001:**
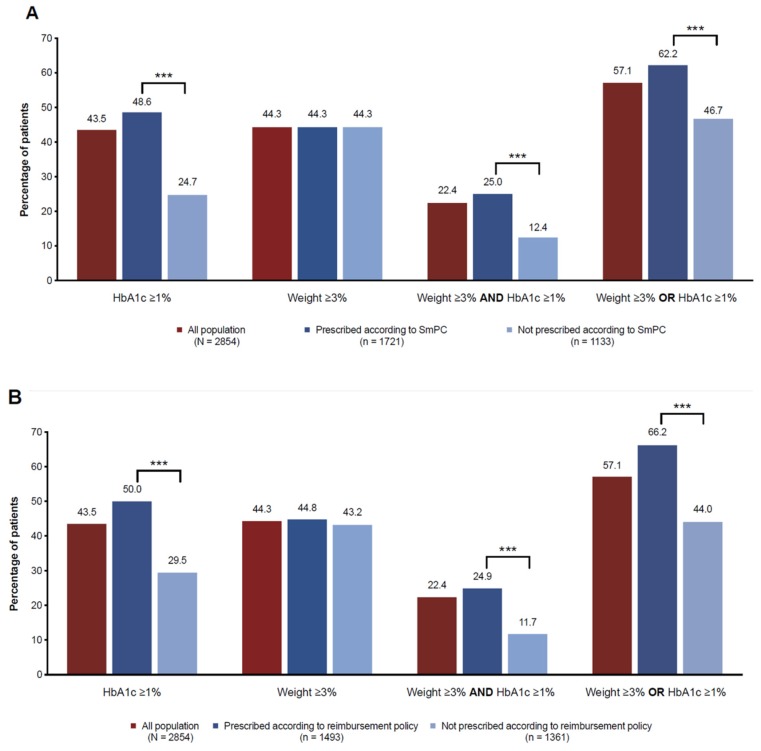
Proportion of patients achieving a beneficial outcome (i.e., HbA1c reduction ≥1% and/or weight reduction ≥3%) after six months of glucagonlike peptide-1 (GLP-1) receptor agonist initiation according to (**A**) the prescription label instructions (SmPC) and to (**B**) the health policy requirements for reimbursement. SmPC, summary of product characteristics; *** *p* < 0.0001.

**Figure 2 jcm-08-01389-f002:**
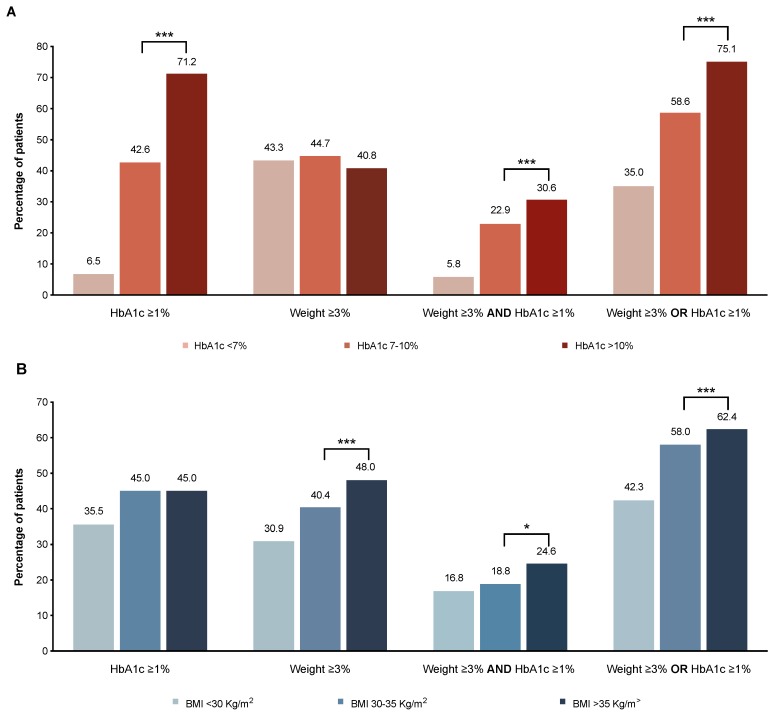
Proportion of patients achieving a beneficial outcome after six months of glucagonlike peptide-1 (GLP-1) receptor agonist initiation according (**A**) to the baseline HbA1c level, and (**B**) to the baseline BMI level. *** *p* < 0.0001; ** *p* < 0.001; * *p* < 0.05.

**Table 1 jcm-08-01389-t001:** Baseline characteristics of the studied cohort.

Characteristic	N	
Age, mean (SD), years	2854	59.5 (9.8)
Gender (male)	2854	50.6%
Diabetes duration, mean (SD), years		9.2 (8.6)
<5 years	705	24.7%
5–10 years	950	33.3%
>10 years	1199	42.0%
Weight, mean (SD), kg	2296	99.7 (18.5)
BMI, mean (SD), kg/m^2^	2304	37.1 (5.9)
HbA1c, %, mean (SD) [mmol/mol]	2206	8.74 (1.6). (72.0 (17.5))
Total Cholesterol, mean (SD), mg/dL	1909	180 (37.6)
HDL-cholesterol, mean (SD), mg/dL	1909	44.9 (11.2)
LDL-cholesterol, mean (SD), mg/dL	1909	99.6 (32.2)
Triglycerides, mean (SD), mg/dL	2133	218 (179)
Heart rate, mean (SD), bpm	2105	80 (12.6)
SBP, mean (SD), mmHg	2456	135 (15.1)
DBP, mean (SD), mmHg	2456	78.3 (9.9)
eGFR, mean (SD), mL/min/1.73 m^2^	2153	85.7 (20.4)
eGFR < 30 mL/min/1.73 m^2^	10	0.5%
eGFR 30–44 mL/min/1.73 m^2^	86	4.0%
eGFR 45–59 mL/min/1.73 m^2^	184	8.5%
eGFR ≥ 60 mL/min/1.73 m^2^	1873	87.0%
Complications, *n* (%)		
Peripheral artery disease	180	6.3%
Ischemic heart disease	420	14.7%
Heart failure	185	6.5%
Stroke	177	6.2%
AD treatment		
No pharmacologic treatment	95	3.3%
1 agent (excluding insulin)	1246	43.7%
1 agent (including insulin)	1454	50.9%
≥2 agents (including insulin)	1305	45.7%
AD treatment by drug class		
Insulin	1483	52.0%
Metformin	2316	81.1%
Sulphonylureas	777	27.2%
DPP4i	749	26.2%
GLP-1RA prescribed		
Liraglutide	1575	56.0%
Exenatide *	668	23.8%
Lixisenatide	569	20.2%

AD, antidiabetic; BMI, body mass index; DBP, diastolic blood pressure; DPP4i, dipeptidyl peptidase-4 (DPP-4) inhibitors; eGFR, estimated glomerular filtration rate using the CDK-Epi equation; GLP-1RA, glucagonlike peptide-1 (GLP-1) receptor agonist; HbA1c, glycated haemoglobin A1c; SBP, systolic blood pressure; SD, standard deviation. * This includes exenatide twice a day and once a week.

**Table 2 jcm-08-01389-t002:** Mean change (SD) in clinical variables from baseline to three to six and 6–12 months of glucagonlike peptide-1 (GLP-1) receptor agonist treatment initiation and according to prescription recommendations or local health policy for reimbursement.

Clinical Variable	Endpoint	Overall Mean Change (SD) *n* = 2251	Prescribed According to SmPC? *			Prescribed According to Reimbursement Policy? **		
			Yes *n* = 1721	No *n* = 1133	*p*-Value	Yes *n* = 1493	No *n* = 1361	*p*-Value
**HbA1c, %**	6 months	−0.84 (1.66)	−0.99 (1.68)	−0.28 (1.46)	<0.001	−1.02 (1.70)	−0.46 (1.51)	<0.001
	12 months	−0.88 (1.57)	−1.03 (1.59)	−0.36 (1.39)	<0.001	−1.03 (1.61)	−0.57 (1.45)	<0.001
BMI, kg/m^2^	6 months	−1.01 (2.36)	−0.99 (2.25)	−1.04 (2.55)	0.674	−1.08 (2.23)	−0.87 (2.59)	0.062
	12 months	−1.08 (2.22)	−1.07 (2.15)	−1.10 (2.35)	0.826	−1.17 (2.08)	−0.89 (2.48)	**0.035**
Weight, kg	6 months	−2.73 (6.44)	−2.67 (6.17)	−2.85 (6.90)	0.559	−2.87 (6.01)	−2.47 (7.19)	0.212
	12 months	−2.86 (5.99)	−2.79 (5.86)	−2.97 (6.25)	0.598	−3.06 (5.57)	−2.44 (6.77)	0.081
Total-Cholesterol, mg/dL	6 months	−6.86 (35.9)	−7.59 (35.6)	−4.13 (36.9)	0.129	−8.05 (36.5)	−4.31 (34.6)	0.052
	12 months	−7.64 (35.3)	−8.31 (35.0)	−5.32 (36.6)	0.263	−9.11 (35.5)	−4.57 (34.8)	0.049
HDL-Cholesterol, mg/dL	6 months	0.09 (7.14)	0.12 (7.01)	−0.02 (7.58)	0.764	0.14 (7.02)	−0.02 (7.38)	0.696
	12 months	−0.05 (7.10)	0.01 (6.96)	−0.27 (7.56)	0.612	−0.03 (7.13)	−0.11 (7.04)	0.859
LDL-Cholesterol, mg/dL	6 months	−5.07 (31.2)	−5.60 (30.8)	−3.08 (32.8)	0.212	−5.65 (31.6)	−3.83 (30.4)	0.278
	12 months	−5.83 (30.5)	−6.21 (29.9)	−4.51 (32.6)	0.471	−6.42 (30.8)	−4.60 (30.0)	0.360
Triglycerides, mg/dL	6 months	−16.42 (166)	−20.52 (181)	−1.28 (90.3)	0.004	−22.41 (174)	−4.50 (149)	0.022
	12 months	−21.30 (156)	−27.16 (170)	−1.29 (91.6)	0.001	−29.41 (163)	−5.59 (141)	0.006
SBP, mmHg	6 months	−0.90 (17.6)	−0.77 (17.3)	−1.11 (18.1)	0.659	−1.01 (17.4)	−0.73 (17.7)	0.710
	12 months	−1.97 (15.7)	−1.92 (15.9)	−2.05 (15.3)	0.864	−2.05 (16.1)	−1.83 (15.1)	0.777
DBP, mmHg	6 months	−0.33 (10.5)	−0.32 (10.1)	−0.36 (11.1)	0.938	−0.42 (10.2)	−0.21 (10.9)	0.643
	12 months	−1.07 (9.46)	−1.05 (9.15)	−1.11 (9.98)	0.917	−1.11 (9.24)	−1.01 (9.80)	0.839
Heart rate, bpm	6 months	0.68 (12.8)	0.58 (12.4)	0.86 (13.5)	0.664	0.64 (12.4)	0.73 (13.4)	0.897
	12 months	0.55 (11.6)	0.26 (11.1)	1.08 (12.6)	0.250	0.36 (11.1)	0.87 (12.5)	0.466
eGFR, mL/min/1.73 m^2^	6 months	−1.63 (11.1)	−1.86 (10.8)	−0.65 (12.1)	0.083	−1.70 (11.0)	−1.48 (11.4)	0.689
	12 months	−1.19 (10.5)	−1.38 (10.5)	−0.41 (10.7)	0.169	−1.16 (10.6)	−1.24 (10.3)	0.884

* GLP-1RAs must be given as add-on therapy in patients with inadequate glycemic control (HbA1c ≥ 7%) with other antidiabetics and in subjects poorly controlled (HbA1c ≥ 7%) and eGFR ≥ 30 mL/min/1.73 m^2.^. ** GLP-1RAs must be given in combination with other antidiabetics and in subjects poorly controlled (HbA1c ≥ 7%) and with a BMI of ≥30 kg/m^2^. BMI, body mass index; DBP, diastolic blood pressure; eGFR, estimated glomerular filtration rate using the CDK-Epi equation; HbA1c, glycated haemoglobin A1c; SBP, systolic blood pressure; SD, standard deviation; SmPC, summary of product characteristics.

**Table 3 jcm-08-01389-t003:** Odds ratio (95% CI) of the variables identified in the multivariate analyses as predictors of clinically beneficial goals after six and 6–12 months of GLP-1RA treatment initiation.

Clinical Variable	After 6 Months				After 6–12 Months			
	Reduction in HbA1c ≥ 1%	Reduction in Weight ≥ 3%	Reduction in HbA1c ≥ 1% AND Weight ≥ 3%	Reduction in HbA1c ≥ 1%, Weight ≥ 3%OR Both	Reduction in HbA1c ≥ 1%	Reduction in Weight ≥ 3%	Reduction in HbA1c ≥ 1% AND Weight ≥ 3%	Reduction in HbA1c ≥ 1%, Weight ≥ 3%OR Both
**Male gender**	–	–	–	–	–	0.73 (0.54–0.97) *p* = 0.036	–	–
**T2D duration** **(10–20 years)**	0.64 (0.43–0.97) *p* = 0.036	–	–	0.67 (0.46–0.96) *p* = 0.031	–	–	2.81* (1.02-7.61) *p* = 0.042	–
**HbA1c**	2.06 (1.83–2.33) *p* < 0.001	–	1.25 (1.12–1.40) *p* < 0.001	1.41 (1.28–1.56) *p* < 0.001	2.14 (1.85–2.48) *p* < 0.001	–	1.42 (1.22–1.66) *p* < 0.001	1.35 (1.22–1.50) *p* < 0.001
**BMI**	–	1.03 (1.01–1.05) *p* = 0.014	1.03 (1.00–1.06) *p* = 0.032	1.02 (1.00–1.05) *p* = 0.043	–	1.04 (1.02–1.07) *p* = 0.001	1.06 (1.02–1.10) *p* = 0.003	1.03 (1.01–1.06) *p* = 0.005
**DBP**	–	–	–	–	–	–	1.03 (1.01–1.06) *p* = 0.018	–
**eGFR**	–	–	–	–	–	–	1.01 (1.00–1.03) *p* = 0.042	–
**Prescribed according to HPR**	–	–	–	1.60 (1.04-2.45) *p* = 0.031	–	–	–	–
**AUC of the model**	0.78	0.61	0.67	0.68	0.79	0.63	0.74	0.67
	(0.76–0.79)	(0.59–0.62)	(0.65–0.69)	(0.66–0.69)	(0.77–0.80)	(0.61–0.65)	(0.71–0.76)	(0.65–0.68)

* T2D duration > 20 years. BMI, body mass index; DBP, diastolic blood pressure; eGFR, estimated glomerular filtration rate using the CDK-Epi equation; HbA1c, glycated haemoglobin A1c; HPR, health policy requirements for reimbursement

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
