# Peer review of "Glucagon-Like Peptide-1 Receptor Agonists in Patients with Type 2 Diabetes: Prescription According to Reimbursement Constraints and Guideline Recommendations in Catalonia"

_jcm, 2019, doi:10.3390/jcm8091389_

Round 1

Reviewer 1 Report

I think that the authors should comment that the eGFR limitations for some GLP-1RAs (including liraglutide, which the majority of their cohort were prescribed) have now been relaxed.

Author Response

Point:  I think that the authors should comment that the eGFR limitations for some GLP-1RAs (including liraglutide, which the majority of their cohort were prescribed) have now been relaxed.

Response: We appreciate the input given by the Reviewer and the interest expressed. The Reviewer is right that the recommendations in patients with impaired kidney function have changed over time. During the study period (i.e., from 2014 to 2016) daily exenatide, liraglutide and lixisenatide were not recommended for patients with eGFR <30 mL/min. In 2017, this changed for liraglutide, which could be used in patients with eGFR over 15 ml/min, while weekly exenatide was not recommended for patients with eGFR <50 mL/min. Following the Reviewer’s suggestion, we have added a paragraph in the discussion section to comment this issue. Moreover, since in our study 10 subjects (0.4%) had eGFR <30 mL/min and 86 cases (3%) had an eGFR between 30 and 45 mL/min, we have conducted additional analyses comparing the variables in Table 1 at baseline between different subgroups of eGFR. We could not find any significant difference regarding age, gender, diabetes duration, HbA1c, BMI or use of insulin or other NIADs.

The manuscript has been modified accordingly as follows:

Results section; Table 1 (page 4; line 177):

Table 1 now includes the number and percentage of patients with different ranges of baseline eGFR values, namely <30 mL/min, 30-44 mL/min, 45-59 mL/min, and ≥60 mL/min.

Discussion section (page 2; lines 322-329):

“Renal insufficiency is not a contraindication but rather a recommendation not to use these drugs due to lack of experience and this has been changed over time. Initially, these drugs were not recommended for patients with eGFR values below 30 mL/min (daily exenatide, liraglutide and lixisenatide), and with eGFR <50 mL/min for weekly exenatide. Later on, the lower limit for eGFR was changed to 15 mL/min for liraglutide. Therefore, we considered, by internal criteria, eGFR <45 mL/min as a not recommended prescription. In addition, in our cohort, only 10 (0.5%) patients initiated GLP-1RA therapy in spite of having a compromised renal function (eGFR <30 mL/min/1.73 m2), a proportion higher than the 9.3% reported by a study conducted in primary care in the US [41].”  

Reviewer 2 Report

The paper is quite interesting, but the novelty is limited. 

In addition, I think that the journal is not suitable for the presented study.

A journal mainly including topics about economical aspects and reimbursement constraints would be more appropriate for publishing the study, assuring a larger audience. 

Author Response

Point: The paper is quite interesting, but the novelty is limited. In addition, I think that the journal is not suitable for the presented study. A journal mainly including topics about economical aspects and reimbursement constraints would be more appropriate for publishing the study, assuring a larger audience.

Response: Thank you for the opportunity to review this manuscript, we appreciate the reviewer’s comment. However, it should be noted that JCM has already published several articles with a similar approach to diabetes therapy as the one of our manuscript. Therefore, we leave the decision to the Editor. Of course, we will accept the final decision adopted.

Reviewer 3 Report

Overall, I think this retrospective analysis can be a valuable addition to the literature, as it investigates a topic that holds important implications for type 2 diabetes mellitus treatment.

However, I have several concerns about how the methodology and results are presented.

Inclusion criteria are not clearly defined.

The study enrolled patients aged 31 to 90 years with an ICD-10 diagnose code for non-insulin-dependent diabetes mellitus (E11) or unspecified diabetes mellitus (E14). However, there are no data how many patients were diagnosed with E14 and why those patients were enrolled in the study with the final diagnosis of type 2 diabetes mellitus?

Moreover, except the general characteristics of cardiovascular complications observed in the study participants (including peripheral artery disease, ischemic heart disease, heart failure, stroke), there are no available data (diagnostic codes) on the incidence of other serious health conditions (i.e. abnormal thyroid function, gastrointestinal disorders, malnutrition, chronic inflammations, neoplasms, and others) that influence the body weight. Therefore, since the electronic database is very general and not personalized, such precise exclusion criteria are necessary to avoid a possible interpretational bias.

Unfortunately, the results of the study are not new, since independent likelihood of achieving a beneficial change in glycaemia and weight loss has been previously observed in many studies with GLP-1RAs usage.

Since there are no currently available data on the off-label use or medication errors with GLP-1RAs (20.5% of patients in the study), the paper could highly benefit from the detailed analysis of this subgroup of patients.

Therefore, after re-defining of study group according to the precise exclusion criteria, please, consider re-working your discussion to contain essential details of how you interpret the novel findings from your study (off-label use or medication errors with GLP-1RAs) and how they might be of clinical relevance.

Author Response

Point 1: Inclusion criteria are not clearly defined.

The study enrolled patients aged 31 to 90 years with an ICD-10 diagnose code for non-insulin-dependent diabetes mellitus (E11) or unspecified diabetes mellitus (E14). However, there are no data on how many patients were diagnosed with E14 and why those patients were enrolled in the study with the final diagnosis of type 2 diabetes mellitus?

Response 1: We agree with the reviewer that E14 codes could be a source of confusion, and this is why we have removed them from the manuscript. Actually, there were no patients with E14 codes because the software does not allow the prescription of GLP-1RAs if the patient does not have a code for DM2 (E11). We have revised this paragraph in the methods section and modified the text as follows (Page 03; Lines 107-108):

“The study enrolled patients aged 31 to 90 years with an E11 ICD-10 (International Classification of Diseases) diagnose code for T2D (i.e., E.11 or E11.0-E11.9).”  

Point 2: Moreover, except the general characteristics of cardiovascular complications observed in the study participants (including peripheral artery disease, ischemic heart disease, heart failure, stroke), there are no available data (diagnostic codes) on the incidence of other serious health conditions (i.e. abnormal thyroid function, gastrointestinal disorders, malnutrition, chronic inflammations, neoplasms, and others) that influence the body weight. Therefore, since the electronic database is very general and not personalized, such precise exclusion criteria are necessary to avoid possible interpretational bias.

Response 2: We thank the reviewer on this comment. Unfortunately, the initial extraction of the data did not include some of these codes. At this stage of the study, we cannot request additional data extraction because of the data anonymization. However, we have reported few cases of pancreas (n=5) and thyroid neoplasms (n=2) as well as pancreatitis (n=11) during the follow-up, which are shown in Supplementary Table S4. Since these could be reasons for treatment discontinuation during the study, we have included this as another limitation of our study in the Discussion section (Page 2; Lines: 374-377):

“Moreover, we did not record the occurrence of some serious health conditions such as abnormal thyroid function, gastrointestinal disorders, malnutrition, or chronic inflammations, among others, which could have influenced BMI or be as well a cause of discontinuation.”

Point 3: Unfortunately, the results of the study are not new, since the independent likelihood of achieving a beneficial change in glycemia and weight loss has been previously observed in many studies with GLP-1RAs usage.

Response 3: We appreciate the comment of the reviewer. However, our study represents real-world data and one of the strengths of such studies is the large sample size. In general, there is a need for real-world data studies in different countries or types of patients because the therapeutic response can be different in different populations. In any case, our study adds information from a Mediterranean population that confirms the results found in other populations.

Point 4: Since there is no currently available data on the off-label use of medication errors with GLP-1RAs (20.5% of patients in the study), the paper could highly benefit from the detailed analysis of this subgroup of patients. Therefore, after re-defining of study group according to the precise exclusion criteria, please, consider re-working your discussion to contain essential details of how you interpret the novel findings from your study (off-label use or medication errors with GLP-1RAs) and how they might be of clinical relevance.

Response 4: We thank the reviewer for raising this point. Actually, the term “off-label” was incorrectly used in the manuscript and could cause misunderstandings. In fact, all patients included in the study had the prescription of the GLP-1RAs as a treatment for type 2 diabetes; therefore, there was no “off label use” of the medication. In addition, in our country, the prescription of these drugs in non-obese type 2 diabetes is not reimbursed by the National Health Service. Moreover, according to the SmPC, these drugs are not recommended in patients with HbA1c <7%, as monotherapy, or with severe/moderate chronic kidney disease. In our study, a total of 376 cases (17.9%) did not accomplish all the criteria recommended by the SmPC, so they could be considered as inappropriate (but not off label). In the manuscript, we conducted specific analyses in this group of patients. In Table 2 and Figure 1, the reader may note that the reductions in HbA1c and BMI were slightly lower in these patients. As expected, there were no differences in age or BMI, but the baseline HbA1c and duration of DM2 were lower (7.54% vs 9.07% and 8.5 vs 9.6 years, respectively).Probably, these were mainly patients with acceptable glycemic control (HbA1c <7%) and the main reason for GLP-1RAs prescription was presumably weight reduction. This is already stated in the discussion section. Please, note that we have changed the term off-label for inappropriate in the main text.

Round 2
